# The Role of Al_4_C_3_ Morphology in Tensile Properties of Carbon Fiber Reinforced 2024 Aluminum Alloy during Thermal Exposure

**DOI:** 10.3390/ma15248828

**Published:** 2022-12-10

**Authors:** Mu Yuan, Jinhao Wu, Qingnan Meng, Chi Zhang, Xinyue Mao, Shiyin Huang, Sifan Wang

**Affiliations:** 1Key Lab of Ministry of Natural Resources for Drilling and Exploitation Technology in Complex Conditions, College of Construction Engineering, Jilin University, Changchun 130026, China; 2College of Mechanical and Electrical Engineering, Jiaxing Nanhu University, Jiaxing 314001, China; 3State Key Laboratory of Superhard Materials, Jilin University, Changchun 130026, China; 4Changchun Institute of Applied Chemistry Chinese Academy of Sciences, Changchun 130026, China

**Keywords:** metal-matrix composites (MMCs), high-temperature properties, mechanical testing, extrusion, interfaces

## Abstract

The aluminum alloy drill pipe suffers long-term high-temperature conditions during ultra-deep well drilling. In this paper, the samples were prepared by vacuum hot pressing, followed by hot extrusion and T6 heat treatment. The mechanical properties of short carbon fiber reinforced 2024 aluminum alloy composites (SCFs/2024 Al) and the microstructure evolution at the interface region thermal exposure at 160 °C for 500 h are discussed. The experimental results showed that the effect of short carbon fiber on 2024 aluminum alloy remained steady throughout the whole process of the heat exposure experiment. The distribution and volume of interface products (Al_4_C_3_) changed with the prolonging of heat exposure time, and connected after coarsening. The evolution of the morphology of Al_4_C_3_ relieved the stress of the interface between carbon fiber and aluminum alloy matrix and enhanced the mechanical properties of the composite.

## 1. Introduction

2024 aluminum alloy (2024 Al) is widely used in aerospace and deep drilling. These industries have certain requirements for the lightweight of parts, which poses a serious challenge to the reliability of related structural materials during use [1,2]. 2024 Al has a low density and high strength [3], good processability, high damage tolerance [4], and excellent fatigue resistance [5] have attracted the attention of many researchers, engineers, and designers. For the extruded 2024 Al, the ultimate tensile strength and yield strength are increased by 131.1 MPa and 60.1 MPa in comparison with the homogenization state [6]. Even small variations in the 2024 Al can lead to a completely different microstructure [7]. Whether aerospace or deep drilling, the materials used are required to withstand high temperatures. At high temperatures, the fine precipitates in the aluminum alloy matrix that determine its strength are coarsened [8]. Thus, the high-temperature performance of 2024 Al is low, which affects its use under high-temperature conditions [9,10].

Carbon fiber has high strength and low density, and its strength maintains at high temperatures [11,12]. Therefore, people add carbon fiber to the aluminum alloy to improve its performance of the aluminum alloy [13]. Li et al. [14,15] added T300 carbon fiber to pure aluminum for increasing the tensile strength of pure aluminum to 200 MPa, with an increased rate of 270%. Lu et al. [16] found the addition of T300 carbon fiber increases the tensile strength of pure aluminum from 59.1 MPa to 144.9 MPa, increasing by 143%. Friler et al. [17] add T300 carbon fiber to the Al-Si matrix increasing the tensile strength of the alloy to 172 MPa, with an increased rate of 53.6%. Imai et al. [18] found when 0.1 vol.% XN-05C carbon fiber was added to 2017 Al the strength of the composite material reached the highest. Singh and Balasubramanian [19] using 6061 Al and XA-S carbon fibers as raw materials, found that the tensile strength of the composite reaches the highest when the mass fraction of carbon fiber is 4 wt.%, and the tensile strength is 420 MPa. Compared with the matrix strength of 300 MPa, it increased by 40%. Zhang et al. [20] found that the highest tensile strength is obtained when the volume fraction of carbon fiber in the matrix is 4 vol.%, which is 4.6% higher than that of 2024 Al. The addition of MPCF carbon fiber was beneficial to improve the tensile strength of the as-sintered 2024 Al composites, which increased from 293 MPa to 406 MPa [21]. Javier [22] tested the mechanical properties of carbon fiber-reinforced A413 aluminum alloy and found that the tensile strength of the composite increased by 25%. Through a comparison of different mechanical models, there was a payload transfer from the matrix to the fiber. Amelie [23] found that adding a small amount of Al/Si alloy to the Al matrix can obtain completely dense materials and enhance the thermal–mechanical properties of Al/C composites. Wang [24] found that in the longitudinal tensile test, the final longitudinal fracture was mainly controlled by the fracture mechanism of graphite fiber.

The addition of carbon fiber can not only improve the performance of aluminum alloy materials at room temperature but also at high temperatures. The addition of 4 vol.% carbon fiber increases the tensile strength of 2024 Al by 13.7% and 18.3% when the test temperature is 423 K and 523 K, respectively [20]. For 7075 Al, the addition of 12 vol.% carbon fiber increases the tensile strength by 15.6% and 16.5% at 150 and 200 °C, respectively [25]. The results show that the addition of carbon fiber helps to improve the performance of aluminum alloy materials under short-term thermal exposure conditions (10 min for Ref. [19,21]). After prolonged thermal exposure, it is worth investigating whether carbon fibers can enhance the performance of the aluminum alloy matrix.

With the development of research on carbon fiber-reinforced aluminum composites, it has been noticed that the interface between carbon fiber and aluminum alloy matrix affects the reinforcement effect of carbon fiber in the aluminum alloy matrix. However, the role of interfacial production Al_4_C_3_ is controversial. One of the parties believes that carbon fiber could be tightly locked in the Al matrix through the anchor effect of Al_4_C_3_, which provides the ability to prevent interface slippage and promotes a noticeably enhanced load transfer in the composites [26,27,28,29]. The other party believes that the formation of brittle Al_4_C_3_ is detrimental to the mechanical properties of carbon materials/Al composites [30,31,32,33]. Metallizing the surface of carbon fiber can effectively reduce the diffusion of carbon elements to the matrix during sintering [34]. The results show that the addition of nickel coating is beneficial to prevent the debonding at the interface between the coating/fiber and the alloy and obtain a finer microstructure. In the presence of nickel coating, the number of grains increases significantly, and the average grain size decreases, which can improve the properties of the resulting composites [35].

In this paper, carbon fibers are added to the 2024 aluminum alloy to improve its strength of 2024 Al. The mechanical properties and microscopic characterization of fiber-reinforced 2024 Al composites after thermal exposure up to 500 h under 160 °C which is the maximum operating temperature to 2024 Al were studied. The change of the interface product after thermal exposure and its influence on tensile strengths were observed and discussed. It is helpful to study the strength prediction of SCFs/2024 Al under long-term high-temperature conditions.

## 2. Materials and Methods

2024 aluminum alloy was selected as the matrix alloy. The powder (Chaowei Nanotechnology Co. Ltd., Shanghai, China) was ~20 µm in D50 particle size. The composition of the raw powder is shown in Table 1. The PAN-based carbon fiber (T700) was provided by Toray Co. Ltd., Tokyo. The PAN-based carbon fiber was immersed in acetone for 4 h to remove the sizing agent and then turned into short carbon fibers (SCFs) by ball milling. Then SCFs were uniformly dispersed in 2024 Al powder through manual mixing using ethyl alcohol. The blended powder was placed in a graphite mold with Φ30 mm diameter and vacuum dried at 90 °C for 12 h to remove ethyl alcohol thoroughly. SCFs reinforced 2024 Al matrix composites were hot pressed fabricated by heating in a vacuum furnace at 580 °C for 40 min, during which a 30 MPa pressure was applied to the powders. Before the extrusion, the SCFs/2024 Al were re-heated at 500 °C for 4 h to remove the surface remains and residual stress. Then, the SCFs/2024 Al composite was extruded at 460 °C with a steady pressure of 500 MPa. The T6 heat treatment process was used for the short carbon fiber-reinforced 2024 aluminum composites prepared in this experiment. First, heat the furnace to 495 °C, then put the sample into the solution temperature and keep it for 60 min. Second, quickly take out the sample and pour it into distilled water to complete the quenching. Within 30 min after quenching, finally, the samples were transferred to a furnace at 190 °C and kept for 12 h to complete the aging treatment. The comparative samples 2024 Al were prepared by using the same method. The samples were cut by the electrical discharge wire-cutting to a specific shape for the following tests, and then the sample was carefully polished. The experimental flow diagram is shown in Figure 1.

The specimens for the tensile test were prepared by ASTM Standard E-8/E8M-09 with a 10 mm gauge length. Tensile data were in situ measured at ambient conditions of 160 °C after 0 h, 0.3 h, 0.5 h, 1 h, 100 h, 200 h, 300 h, and 500 h thermal exposure, respectively. Each data point is tested three times, and the average value is taken. The Vickers hardness was measured after cooling to room temperature via a microhardness tester (HXD-1000TM, Shanghai, China) under an applied load of 100 g for 15 s. At least seven measurements were performed to ensure the accuracy of the results. The phase constituents of samples were identified by X-ray diffraction (XRD, DX2700, Dandong, China) using Cu Kα radiation in step mode from 20° to 80°. The calculation of average crystallite size was based on XRD patterns. A scanning electron microscope (SEM, S4800, Tokyo, Japan) was used to record the surface topography and fracture morphology. The area just below the fracture was sampled, polished, and observed by transmission electron microscopy (TEM, Titan G2 60–300, Hillsboro, NC, USA) equipped with scanning transmission electron microscopy (STEM) and energy dispersive spectrometer (EDS, Super X).

## 3. Results

The optical morphology for the SCFs/2024 Al is presented in Figure 2. As shown in Figure 2, SCFs are dispersed uniformly in the 2024 Al matrix with no obvious clustering being observed. Similar results were obtained in our previous research in SCFs/2024 Al and SCFs/7075 Al [20,25]. In the transverse section, the shape of carbon fiber is mostly round. Therefore, the orientation of carbon fiber inside the matrix agrees with the orientation of hot extrusion. This directional structure also helps to improve the thermal conductivity of the material [36,37]. The average length of carbon fiber is 37.15 μm.

Figure 3a presents the XRD patterns for 2024 Al which had exposure at 160 °C with time ranging from 0 h to 500 h. According to the XRD results, major aluminum peaks are detected at 38.472° (1 1 1), 44.738° (2 0 0), 65.133° (2 2 0), and 78.227° (3 1 1) (PDF#04–0787). Figure 3b shows the red dashed line in Figure 3a. As shown in Figure 3b, it is worth noting that there are peaks representing Al_2_CuMg when the time of heat-treated got 100 h. Al_2_CuMg peaks are detected at 27.249° (1 1 1), 35.008° (1 1 2), 38.940° (1 3 1), and 40.99° (0 4 1) (PDF#28-0014). Figure 3c shows the blue dashed line in Figure 3a. In Figure 3c the XRD data with thermal exposure times of 0 h, 0.3 h, 0.5 h, 1 h, and 500 h were selected as representative presentations. CuAl_2_ which is detected at 29.385° (2 0 0), 47.331° (3 1 0), and 47.807° (2 0 2) (PDF#25-0012) can be detected regardless of the thermal exposure time. Figure 4a presents the XRD patterns for SCFs/2024 Al composite which had exposure at 160 °C with time ranging from 0 h to 500 h. Figure 4b–d regions show the red, blue, and green regions in Figure 4a, respectively. Figure 4a–c show similar results to Figure 3a–c meaning that the influence of addition SCFs on the phase structure of the matrix is slight. Meanwhile, when the thermal exposure time reaches 300 h, the Al_4_C_3_ peaks detected at 30.774° (1 0 1), 37.850° (1 0 4), and 45.594° (0 0 8) (PDF#50-0740) are found.

From the results of XRD, it can be judged that the grains of Al_2_CuMg and Al_4_C_3_ grow continuously during the increase in thermal exposure time because Al_2_CuMg and Al_4_C_3_ cannot be found by XRD when the grain size is small. The aluminum grain size was calculated by Scherrer’s equation [38]
(1)$$D=\frac{K\lambda }{\beta \cos\theta }$$
where *D* is grain size; *β* is the integral width; *K* takes 1.0; λ is the X-ray wavelength; *θ* is the diffraction angle. As shown in Figure 5, the aluminum grain size of SCFs/2024 Al and 2024 Al is consistent within the tolerance range. The grain size showed a slight upward trend in both of SCFs/2024 Al and 2024 Al. In Figure 5, the abscissa is processed by the lg function change. Samples without thermal exposure are defined as 0.1 h. Later in this article, when there is thermal exposure time as the abscissa, the same treatment as Figure 5 is adopted.

Figure 6 presents the Vickers hardness results of SCFs/2024 Al and 2024 Al which had exposure at 160 °C with time ranging from 0 h to 500 h (Take the lg function on the abscissa). For 2024 Al, the hardness drop is small when the thermal exposure time is no more than 1 h, from HV 120 to HV 104, the hardness is reduced by 13.3%. The hardness is stable at HV 92 when the thermal exposure time is over 100 h. Compared to the initial hardness (HV 120), the hardness was reduced by 23%. For SCFs/2024 Al, the hardness decreased from HV 140 to HV 131 when the thermal exposure time was no more than 1 h, reduced by 6.4%. The hardness is stable at HV 113 when the thermal exposure time is over 100 h. Compared to the initial hardness (HV 140), the hardness was reduced by 19%. The hardness for 2024 Al is always lower than SCFs/2024 Al. The addition of SCFs increased the hardness from HV 120 to HV 140, enhanced by 16.7%, when the thermal exposure time is 0 h. When the thermal exposure time was more than 100 h, the addition of SCFs increased the hardness from HV 92 to HV 113, enhanced by 21.5%. Because the indentation would cover SCF in SCFs/2024 Al SCF can enhance the hardness and modulus of the nearby matrix [20]. Whatever the 2024 Al and SCFs/2024 Al, HV is in a decreasing tendency. As the thermal exposure time increases, the aluminum and Al_2_CuMg grain sizes would increase, resulting in a decrease in hardness.

Figure 7 shows the tensile strengths (TS) for 2024 Al and SCFs/2024 Al with the growth of thermal exposure time. For 2024 Al, TS decreases from 384.0 MPa to 229.5 MPa, reduced by 40.2%. Meanwhile, for SCFs/2024 Al, TS decreases from 460.3 MPa to 295.9 MPa, reduced by 35.7%. The TS for 2024 Al is always lower than the TS for SCFs/2024 Al. Contrasting the initial TS between 2024 Al and SCFs/2024 Al, the addition of SCFs helps the 2024 Al to improve the TS from 384.0 MPa to 460.3 MPa, rising by 19.9%. When the thermal exposure time extends to 500 h, the addition of SCFs helps the 2024 Al to improve the TS from 229.5 MPa to 295.9 MPa, rising by 28.9%. Therefore, the addition of carbon fiber can reduce the decrease in tensile strength caused by long-time thermal exposure. In addition, the TS fitting curve for 2024 Al leveled off before 0.3 h. In contrast, the TS fitting curve for SCFs/2024 Al leveled off between 0.5 h and 1 h. It means that the addition of SCFs gives rise to an improvement in the stability of performance during thermal exposure.

Figure 8 shows the yield strengths (YS) for 2024 Al and SCFs/2024 Al with the growth of thermal exposure time. For both 2024 Al and SCFs/2024 Al, the YS fitting curve shows a trend of first decreasing and then stable. When the thermal exposure time was over 100 h, YS for 2024 Al and SCFs/2024 Al were stable at 143.6 MPa and 170.1 MPa (reduced by 51.0% and 47.8%), respectively. The YS for 2024 Al is also always lower than the YS for SCFs/2024 Al. Contrasting the initial YS between 2024 Al and SCFs/2024 Al, the addition of SCFs helps the 2024 Al to improve the YS from 293.2 MPa to 326.0 MPa, rising by 11.2%. When the thermal exposure time extends to 500 h, the addition of SCFs helps the 2024 Al to improve the YS from 143.6 MPa to 170.1 MPa, rising by 18.5%. The addition of SCFs is great for improving yield strength especially when the heat-treated time is over 100 h.

SEM images of tensile fracture for SCFs/2024 Al are shown in Figure 9a–d. For the fracture of the sample, lots of cut fibers (marked in red dotted circles) are observed. When the thermal exposure time was less than 0.5 h, a large number of carbon fibers were sheared. This phenomenon is consistent with our previous study that roughened interfacial products lead to adverse effects on carbon fibers [20]. As the thermal exposure time increased to 1 h, the fraction of carbon fibers that were sheared dropped. It can be seen from Figure 9b,c that the proportion of sheared fibers to the total number of fibers in the photo is decreasing. When the thermal exposure was extended to 500 h, only a few carbon fibers were cut. It can be judged that the integrity of the carbon fibers in the aluminum matrix is gradually preserved when the thermal exposure time is covered to 1 h.

From Figure 10, lath-like products can be observed near the carbon fiber interface. This product forms nearby carbon fiber and diffuses into the aluminum matrix. Through the analysis of the EDS element content of the product, it can be seen that its main constituent elements are: C (28 at. %), Al (56 at. %), O (15 at. %) and Mg (1 at. %). The contents of Al and Mg must be overestimated because of the influence of the 2024 Al matrix. The detection of the O element cannot be avoided. Therefore, the main component of the interface products should be an aluminum carbide. Considering the presence of Al_4_C_3_ products observed in the XRD test results and the lath-like morphology, the interface product is attributed to Al_4_C_3_.

Figure 11a–f shows the HAADF STEM morphology characterization at the interface between carbon fiber and Al matrix in SCFs/2024 Al composites (heat-treated 0 h). In addition to the Al matrix and the interface lath-like product Al_4_C_3_, the following products can also be observed in Figure 10. The first one is the lamellar precipitation phase in the Al matrix, which is the common θ’ (CuAl_2_) precipitation phase in the Al-Cu-Mg-based aluminum alloys [39], which also echoes the previous XRD test results. The second one is the precipitates around Al_4_C_3_ and at the Al grain boundaries. From the element distribution of EDS mapping (Figure 11e), it can be seen that this precipitate is an Mn-rich phase. The last one is a relatively large-sized bright white precipitate, which is an Mn-Cu-rich phase.

As shown in Figure 12, the thickness of the Al_4_C_3_ interfacial layer around the carbon fiber is about 0.31 µm in the sample with a thermal exposure time of 1 h. With increasing thermal exposure time to 500 h, the thickness of the Al_4_C_3_ interfacial layer grows to 0.84 µm.

As shown in Figure 13, the carbon fiber surface is relatively smooth when the samples were not exposed to heat. When the thermal exposure time was increased to 1 h, rod-shaped erosion pits were almost fully covered on the surface of carbon fiber. It implies the carbon fiber surface is gradually covered by interfacial product Al_4_C_3_ with increasing thermal exposure time increased to 1 h. Together with Figure 12a, a continual interfacial Al_4_C_3_ layer is formed around carbon fiber. By further increasing the thermal exposure time to 500 h, severe erosion of the carbon fiber surface was observed. Combined with Figure 12b, the interface Al_4_C_3_ layer is growing.

## 4. Discussion

According to the situation of the interface product Al_4_C_3_ around the carbon fiber observed by SEM, a schematic diagram of Figure 14a–d is made.

The lath-like Al_4_C_3_ forms between carbon fiber and matrix during the composite forming process. And the lath-like Al_4_C_3_ coarsens with prolonged thermal exposure time, as shown in Figure 12a and Figure 13b. The existence of lath-like Al_4_C_3_ gives rise to stress concertation resulting in a shear fracture of carbon fiber. As shown in Figure 9a, therefore, lots of carbon fibers are sheared during the tensile test. With increasing thermal exposure time, Al_4_C_3_ continually coarsens and connects, which gives rise to a stronger interface and less stress concentration. Thus, the number of cut fibers decreases with the increase in thermal exposure time, as shown in Figure 9b–d. With further increasing thermal exposure time, Al_4_C_3_ combines into a whole layer, leading to the disappearance of stress concentration. Thereby, almost no fiber is cut after a long time of thermal exposure (longer than 1 h), as shown in Figure 9d. That is, the change of Al_4_C_3_ protects the integrity of SCFs in the aluminum matrix.

To explain how the addition of carbon fiber affects the strength of 2024 aluminum alloy, the strengthening mechanism of SCFs/2024 Al was discussed. The common strengthening modes existing in aluminum alloy composites are fine-grain strengthening and Orowan strengthening [40]. The strengthening effect by grain refinement (Δσ_GR_) can be calculated by the following formula [41].
(2)$$\Delta \sigma_{GR}=K\left(D_{c}^{-0.5}-D_{m}^{-0.5}\right)$$
where *Dc* and *Dm* are the grain size of the composite and unreinforced matrix, respectively, and *K* = 0.04 MPa m^1/2^ for aluminum. In aluminum, there will be problems with solubility due to the presence of other elements, and thus precipitate phases will appear. According to Figure 5, the addition of SCFs had a slight influence on the aluminum grain size. Thus, the strengthening effect of grain refinement caused by the addition of SCFs was regarded as 0.

The uniform distribution of precipitated particles in the matrix hinders the movement of dislocations, resulting in a strengthening effect [42]. For example, the interaction between uniformly dispersed nanoparticles and dislocations blocks the movement of dislocations, leading to dislocation accumulation. To achieve the strengthening effect by the Orowan mechanism within the framework of dispersion hardening [43]. This enhancement is called Orowan enhancement. It can be calculated according to the following formula [44].
(3)$$\Delta \tau_{p}=\frac{Gb}{2\pi \sqrt{1-v}}\cdot \frac{1}{\lambda }\cdot \ln\frac{r_{0}}{r_{i}}$$
where *G* is the shear modulus of the aluminum matrix phase, *ν* is Poisson’s ratio, *λ* is the effective planar inter-particle spacing, *r_o_* and *r_i_* represent outer and inner cut-off radii, respectively, for matrix dislocations, and b is the magnitude of the Burgers vector of those dislocations. Orowan strengthening formula shows that the strengthening is mainly related to the diameter and distribution of the second phase particles [45].

Some scholars calculated the Orowan strengthening effect produced by lath-like Al_4_C_3_ uniformly distributed in the matrix, with a value of 29.5 MPa [46]. However, in this experiment, the size of the carbon fibers is larger than that of the precipitated phase which can cause dispersion strengthening. Furthermore, the Al_4_C_3_ are dependent on the surface of carbon fibers, not uniformly dispersed in the matrix. Thus, the Orowan strengthening brought by the addition of carbon fibers can be ignored in SCFs/2024 Al composites.

There are also stress transfer reinforcements dominated by carbon fibers. The value of SCFs aggrandizement can be calculated as [47].
(4)$$\sigma_{c}=\sigma_{F}V_{F}\frac{l}{l_{c}}+\sigma_{M}\left(1-V_{F}\right)\;For\;l<l_{c},$$
where $l_{c}=\frac{\sigma_{F}D}{2\tau_{M}}$, τ*_M_* represents the shear strength of the matrix, which can be obtained from the elastic modulus, $l_{c}$ range from 117 μm to 239 μm, *σ_F_* is the strength of carbon fiber and *σ_M_* is the strength of 2024 Al. *V_F_* is the carbon fiber volume fraction.

The relationship between theoretical strength and actual strength for composites were shown in Figure 15. Theoretical strength is equated with the actual strength of 2024 Al adds the corresponding value of SCFs aggrandizement. As shown in Figure 15a, the two curves are in good agreement. The theoretical SCFs/2024 Al yield strength curve also shows a trend of first decreasing and then stable. In Figure 15b, the theoretical curve of TS is lower than the actual value of TS. This shows that when the composite material is broken, there are other influencing factors to influence the strength of SCFs/2024 Al in addition to the stress transfer effect, compared with the strength of 2024 Al. The calculation model used in calculating the carbon fiber reinforcement value does not account for the effects of Al_4_C_3_. The existence of lath-like Al_4_C_3_ is beneficial to form a firm connection between matrix and carbon fiber because of the anchor effect. Al_4_C_3_ has a relatively smooth and clean contact interface with the aluminum matrix with high bonding strength [48]. The formation of Al_4_C_3_ is conducive to improving the interfacial load transfer capacity [49,50]. In the process of breaking the SCFs/2024 Al, Al_4_C_3_ enables a tight locking of the carbon fiber in place by anchor effects [51]. Thus, anchor effects can increase the overall TS of SCFs/2024 Al upon carbon fiber failure. Li [46] considered that the anchoring effect of interface products was the main reason for the difference between the calculated results and the experimental results when calculating the strengthening effect of CNT on aluminum alloy gold materials. This is consistent with the results of this paper. In addition, the TS turning point of the theoretical curve is before the thermal exposure time of 20 min, while the TS curve turning point of the actual SCFs/2024 Al is between 30 min and 1 h. Referring to the schematic diagram given above, the Al_4_C_3_ are gradually connected to form continuous interfacial products, which weakens the effect of stress concentration. This reduces the number of damaged carbon fibers. Thus, the SCFs/2024 Al tensile strength falls more slowly with increasing thermal exposure time. This indicates that the SCFs/2024 Al exhibits better thermal stability than 2024 Al. The obtained results are consistent with the Al_4_C_3_ change model established in this paper. Compared with the YS, the reason why the theoretical curve of yield strength fits well with the fitted curve of the actual value is that the yielding of the composite material does not involve the failure of carbon fibers. The effect of anchoring effect on the yield stage also does not appear.

In this paper, the anchoring effect and stress concentration effect of Al_4_C_3_ is considered to exist. The rivet effect increases with the increasing thermal exposure time. The stress concentration effect is weakened with the change of the interfacial products.

## 5. Conclusions

Carbon fiber reinforced 2024 Al was produced by vacuum hot pressing, followed by hot extrusion and T6 heat treatment. The addition of carbon fiber gives rise to an increase in tensile strength from 384.0 to 460.3 MPa (increased by 19.9%), and in yield strength from 293.2 to 326.0 MPa (increased by 11.2%). Moreover, the improvement from carbon fiber persisted even after prolonged thermal exposure. After heating at 160 °C for 500 h, the tensile strength of 2024 Al decreases to and stabilizes at 229.5 MPa, in contrast, the addition of carbon fiber results in a delay of strength reduction and keeps the tensile strength at 326.0 MPa. Similarly, after thermal exposure, 2024 Al exhibits a low yield strength (143.6 MPa), and the addition of carbon fibers is a benefit to keep the yield strength to a relatively high yield strength (170.1 MPa).

Through theoretical calculation, the carbon fibers in the 2024 Al matrix enhance the strength of 2024 Al through the effect of stress transfer. In addition, the interfacial production of Al_4_C_3_ exhibits a duality of strengthening and weakening. The anchoring effect of Al_4_C_3_ gives rise to an increase in the strength of alloys. Meanwhile, however, carbon fibers are cut because the lath-like Al_4_C_3_ results in a stress concentration. Therefore, the lath-like Al_4_C_3_ also gives rise to a decrease in the strength of alloys.

With the increase in heat exposure time, the matrix of 2024 Al softens, resulting in a decrease in the strength of SCFs/2024 Al. In contrast, the coarsened Al_4_C_3_ are connected, reducing the stress concentration and avoiding the cut of carbon fibers. Thus, the stress transmission effect of carbon fibers is improved, and the strength turning point of SCFs/2024 Al is delayed.

The addition of carbon fiber can improve the strength of 2024 Al under high temperature environments for a long time. Therefore, SCFs/2024 Al is a potential material that can be used in high-temperature service equipment (such as drill pipe for high-temperature drilling).

## Figures and Tables

**Figure 1 materials-15-08828-f001:**
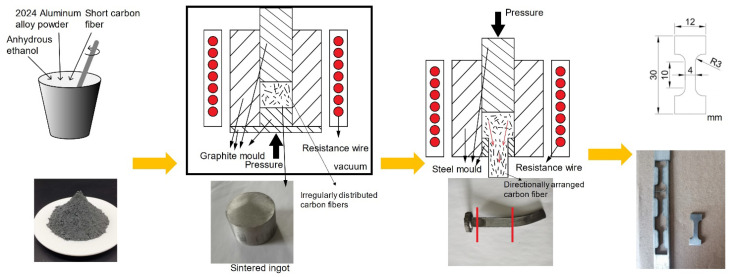
Experimental flow diagram.

**Figure 2 materials-15-08828-f002:**
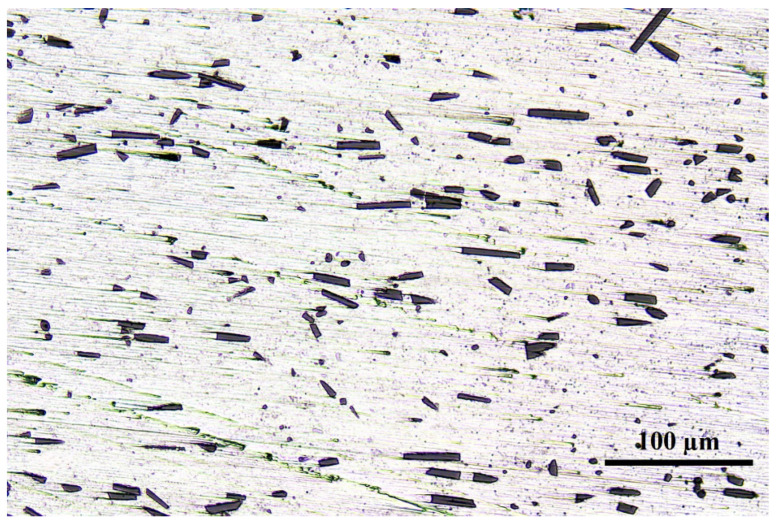
OM image of the surface along the extrusion direction of the sample.

**Figure 3 materials-15-08828-f003:**
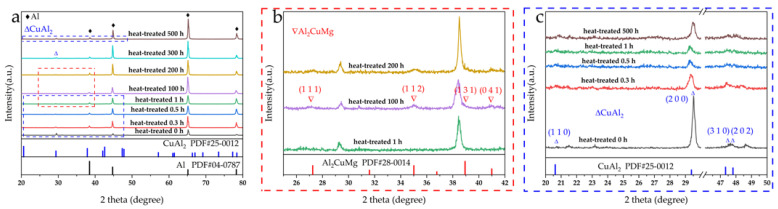
(**a**) XRD patterns of 2024 Al, with heat-treated time, ranged from 0 h to 500 h at 160 °C; (**b**) the red dotted area in (**a**); (**c**) the blue dotted area in (**a**).

**Figure 4 materials-15-08828-f004:**
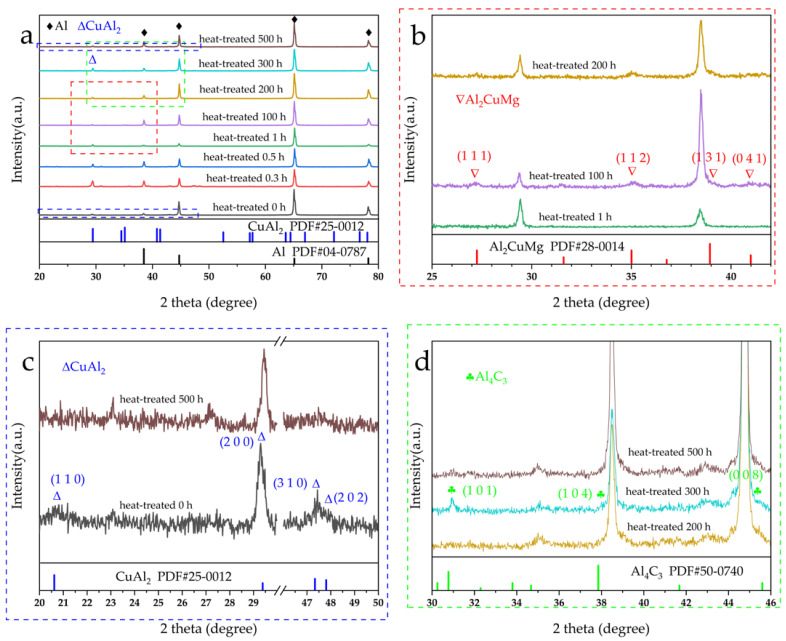
(**a**) XRD patterns of SCFs/2024 Al, with a heat-treated time range from 0 h to 500 h at 160 °C; (**b**) the red dotted area in (**a**); (**c**) the blue dotted area in (**a**); (**d**) the green dotted area in (**a**).

**Figure 5 materials-15-08828-f005:**
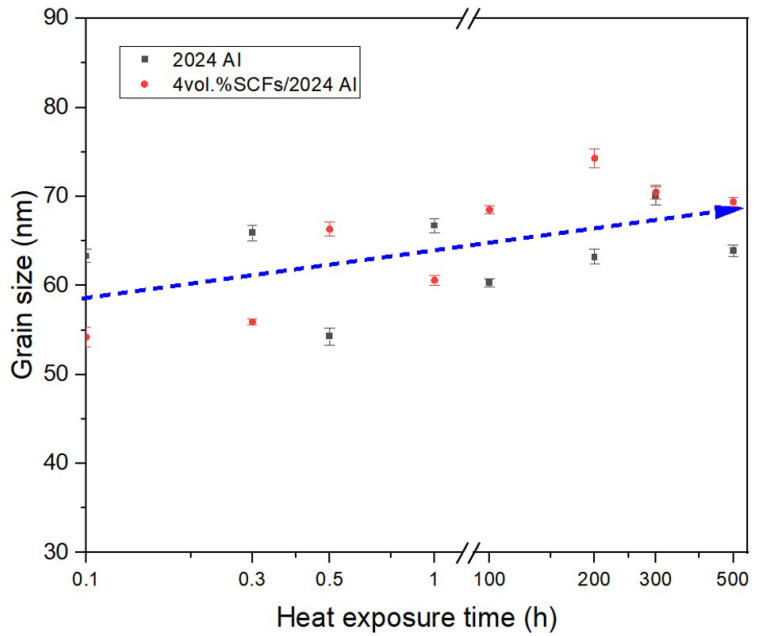
The change of aluminum grain size for both 2024 Al and SCFs/2024 Al with the change of thermal exposure time. (Samples without thermal exposure are defined as 0.1 h).

**Figure 6 materials-15-08828-f006:**
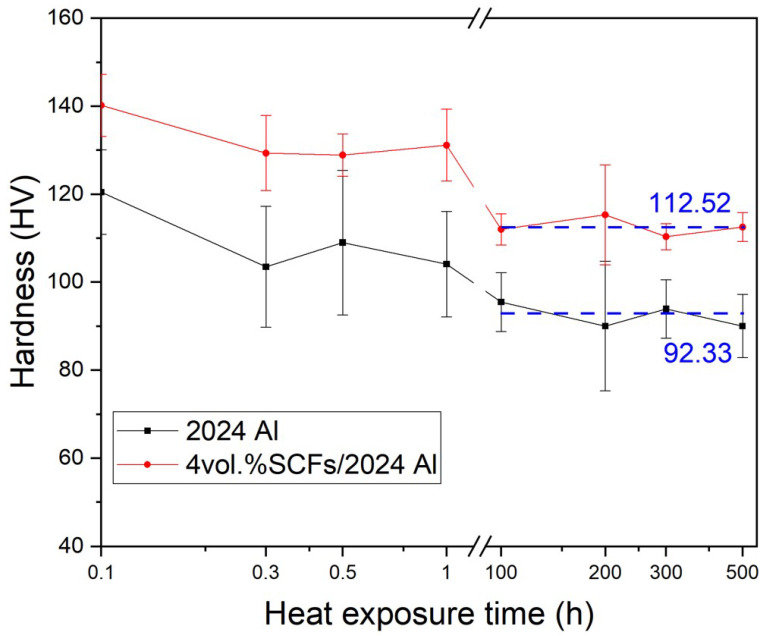
Hardness of 2024 Al and SCFs/2024 Al with the change of thermal exposure time. (Samples without thermal exposure are defined as 0.1 h).

**Figure 7 materials-15-08828-f007:**
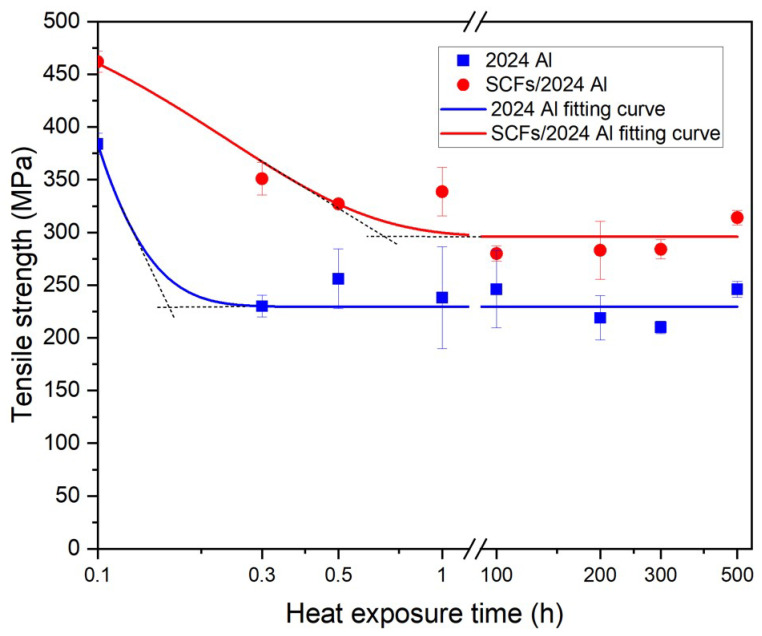
Tensile strength for 2024 Al and SCFs/2024 Al with the change of thermal exposure time. (Samples without thermal exposure are defined as 0.1 h).

**Figure 8 materials-15-08828-f008:**
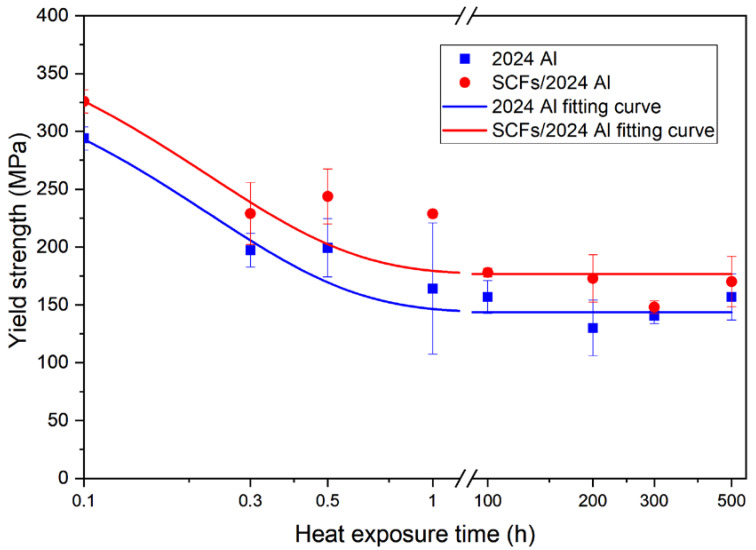
Yield strength of 2024 Al and SCFs/2024 Al with the change of thermal exposure time. (Samples without thermal exposure are defined as 0.1 h).

**Figure 9 materials-15-08828-f009:**
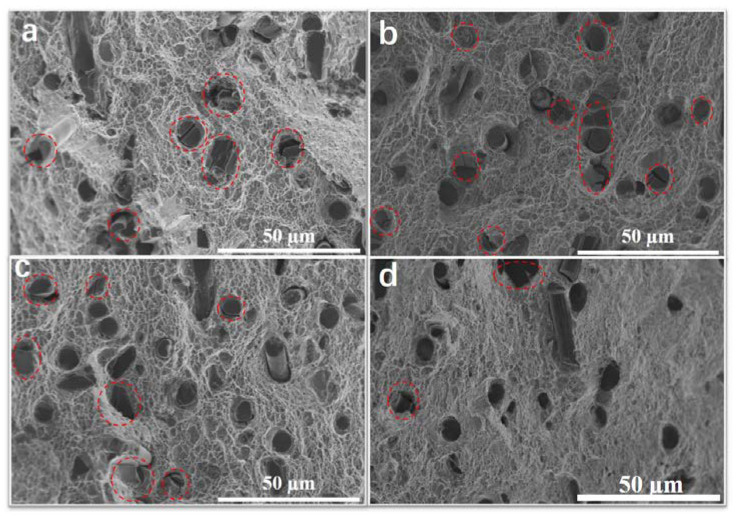
Typical morphologies fracture surface of SCFs/2024 Al. (**a**) thermal exposure time 0 h; (**b**) thermal exposure time 0.5 h; (**c**) thermal exposure time 1 h; (**d**) thermal exposure time 500 h. Red dotted line area is the cut carbon fiber.

**Figure 10 materials-15-08828-f010:**
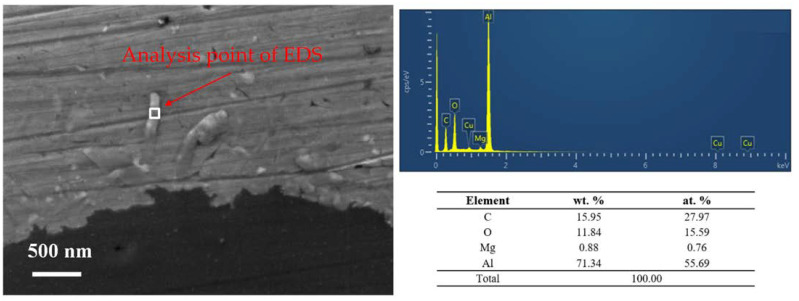
Morphologies fracture surface of SCFs/2024 Al (polished smooth), thermal exposure time 0 h; SEM morphology of the stack-like interface product in SCFs/2024 Al composite and its corresponding elements distribution obtained by EDS analysis.

**Figure 11 materials-15-08828-f011:**
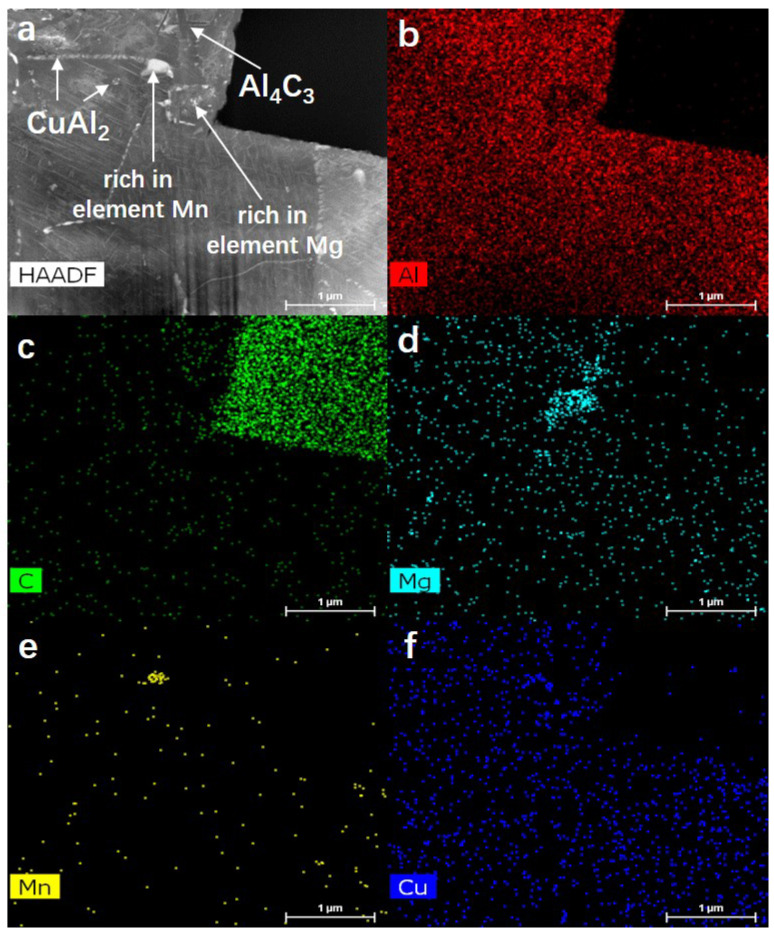
(**a**) HAADF STEM morphology for the interface structure of SCFs/2024 Al composite and the distribution of the corresponding elements for (**b**)-Al, (**c**)-C, (**d**)-Cu, (**e**)-Mg, and (**f**)-Mn obtained by EDS mapping analysis. Thermal exposure time 0 h.

**Figure 12 materials-15-08828-f012:**
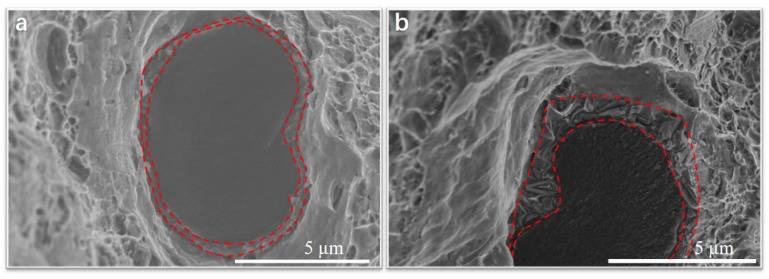
Morphologies fracture surface of SCFs/2024 Al. (**a**) thermal exposure time 1 h; (**b**) thermal exposure time 500 h. Red dotted line area is the Al_4_C_3_ interfacial layer.

**Figure 13 materials-15-08828-f013:**
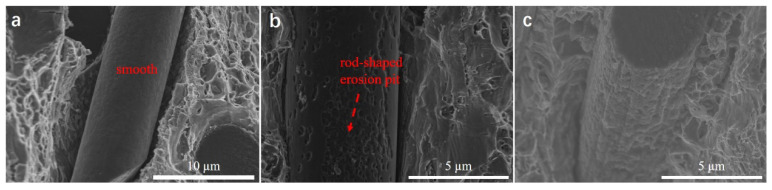
Morphologies fracture surface of SCFs/2024 Al about surface morphology of carbon fiber. (**a**) thermal exposure time 0 h; (**b**) thermal exposure time 1h; (**c**) thermal exposure time 500 h.

**Figure 14 materials-15-08828-f014:**
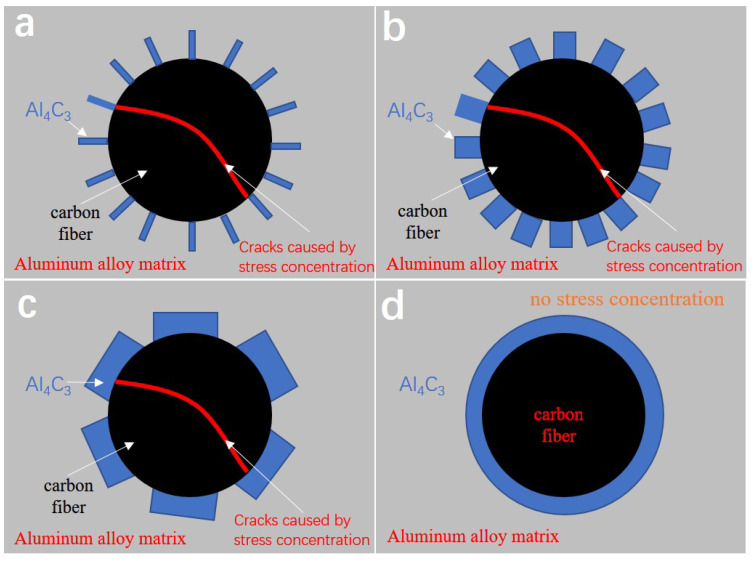
Schematic representation of Al_4_C_3._ (**a**) Initial morphology of Al_4_C_3_; (**b**). Morphology of coarsened Al_4_C_3_; (**c**) Morphology of Al_4_C_3_ connected together; (**d**) Al_4_C_3_ layer completely covering carbon fiber surface.

**Figure 15 materials-15-08828-f015:**
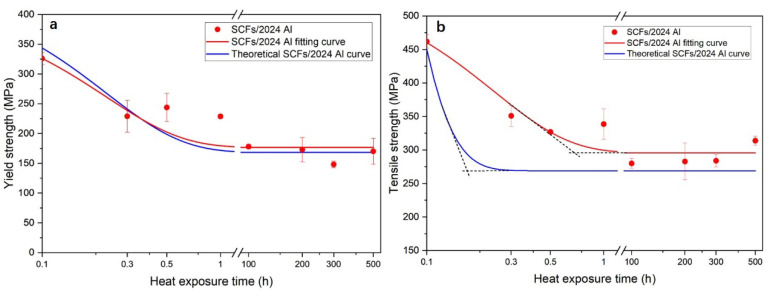
(**a**) Yield strength actual value fitting curve and theoretical curve; (**b**) Tensile strength actual value fitting curve and theoretical curve.

**Table 1 materials-15-08828-t001:** Chemical composition of the as-received 2024 Al powder (wt.%).

Chemical Element	Cu	Mg	Mn	Zn	Fe	Si	Al
Content (wt.%)	3.952	1.283	0.450	0.137	0.125	0.087	Bal.

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
