# Peer review of "The Role of Al4C3 Morphology in Tensile Properties of Carbon Fiber Reinforced 2024 Aluminum Alloy during Thermal Exposure"

_materials, 2022, doi:10.3390/ma15248828_

Round 1

Reviewer 1 Report

I want to congratulate the authors for the manuscript titled as “The role of Al4C3 morphology in tensile properties of carbon  fiber reinforced 2024 aluminum alloy during thermal exposure”. The article is interesting and deserves the attention of readers. However, there are several points in the article that require further explanation.

Abstract: writing is too generalized, and it is too long especially for explanation about the material and method process. The main theme of this paper is not described in the abstract. Abstract section should be concisely reflected the content and summarize the problem, the method, the results, and the conclusions. The abstract needs to be improved. Demonstrate in the abstract novelty, practical significance.

Each one of the cited references  must be discussed individually and demonstrate their significance to your work. Not [3-7], should be [3] text what is presented in the manuscript [3] text what is presented in the manuscript [4].

I propose authors give the actual photos belong to experimental infrastructure. Also, a graphical abstract about experimental setup would be helpful for the presentation of the study.

Fig.2 caption is seemingly wrong. It is not SEM image; it looks like OM image. Please correct it.

The alignment of Fig. 13 is wrong. Please revise it.

The XRD diffraction pattern of Fig.2 and 3 should be marked with a standard PDF card. Also, the referred PDF card No. in XRD results should be supplemented in the corresponding text of Fig. 2 and 3.

Please check the manuscript for wrong choice of words, grammatical errors and incoherent sentence structure. Writers should pay more attention of singular / plural nouns. Also, they should control the spell check/ punctuation of words and sentences. Also, please recheck upper and lower case letter such as Al4C3 especially for manuscript title, abstract, introduction, conclusion, etc.. (such as page 11 line 263 figure 11a should be corrected as Figure). In addition, spaces should be added between words and numbers. The authors can use suitable grammar-checking software / use the help of a native English speaker to correct these mistakes. Please fix the typographical and eventual language problems in paper.

Some of the text in figures are not readable. Improve the sharpness of all low quality images and their text and numbers especially for XRD figs (Fig 2 and 3).

The results section needs to be improved according to using proper citations and support the findings. Please improve all results with 5-6 lines with commenting on the figures.Lack of detailed information about the crystallographic properties (such as grain size, XRD results) and hardness/strengthening, Orowan mechanisms. The fact is that there many works relating that the Al-based MMCs have much better discussion than this work. Please look on paper https://doi.org/10.1016/j.apt.2021.08.031, https://doi.org/10.1002/adem.201400232,  https://doi.org/10.1016/j.jmrt.2021.07.109, and try to add them to discussion.

It is necessary to give quantitative and qualitative indicators of the proposed method in conclusion. Conclusions should be written in more detail adding numeric data. Conclusions section is inadequate. There should be the importance of the study in detail, comparison results with other approaches in literature, the success of the prediction and computational results.

The paper is well-organized yet there is a reference problem. Your reference list contains only one paper from Materials journal. If your work is convenient for this journal's context, then there are many references from this source. Besides, references are not enough. Such a work deserves many citations related with Al-based MMCs. Minimum 10-15 references need to be added and some of them should be discussed.

Reviewer 2 Report

Manuscript ID: materials-2030249

Title: The role of Al4C3 morphology in tensile properties of carbon fiber reinforced 2024 aluminum alloy during thermal exposure

In the current paper, the mechanical properties and microscopic characterization of fiber-reinforced 2024 Al composites after thermal exposure are investigated. The organization of the paper is acceptable. A widely used topic has been investigated and the subject is in the scope of the journal. The following comments should be addressed by the authors:

·         Enrich conclusion section with more quantitative data from the research.

·         The innovation and aims of the current study should be explained in more details at the end of the introduction.

·         According to the experimental tests, has the uncertainty analysis been done?

·         Regarding the subject of the research, it is useful to discuss about the methods to deal with heat conduction in reinforced composite materials. Referring following papers will be useful in this regard:

10.1007/s11630-021-1517-1; 10.1016/j.ijthermalsci.2011.09.002

Round 2

Reviewer 1 Report

The authors revised the paper named “

The role of Al4C3 morphology in tensile properties of carbon fiber reinforced 2024 aluminum alloy during thermal exposure”. I have reviewed the paper and see that the authors have been made many contributions to article. I think the paper is ready for publishing at the final version. My decision is about to accept it.